# Changes in Faecal Microbiota Profile and Plasma Biomarkers following the Administration of an Antioxidant Oleuropein-Rich Leaf Extract in a Rat Model Mimicking Colorectal Cancer

**DOI:** 10.3390/antiox13060724

**Published:** 2024-06-14

**Authors:** Sofia Chioccioli, Gabriele Rocchetti, Jessica Ruzzolini, Silvia Urciuoli, Francesco Vitali, Gianluca Bartolucci, Marco Pallecchi, Giovanna Caderni, Carlotta De Filippo, Chiara Nediani, Luigi Lucini

**Affiliations:** 1Department of Neurosciences, Psychology, Drug Research and Child Health (NEUROFARBA), University of Florence, 50134 Florence, Italy; sofia.chioccioli@unifi.it (S.C.); gianluca.bartolucci@unifi.it (G.B.); giovanna.caderni@unifi.it (G.C.); 2Department of Animal Science, Food and Nutrition, Università Cattolica del Sacro Cuore, Via Emilia Parmense 84, 29122 Piacenza, Italy; 3Department of Experimental and Clinical Biomedical Sciences “Mario Serio”, University of Florence, 50134 Florence, Italy; jessica.ruzzolini@unifi.it; 4PhytoLab (Pharmaceutical, Cosmetic, Food Supplement Technology and Analysis)-DiSIA, Department of Statistics, Informatics, Applications “Giuseppe Parenti”, Scientific and Technological Pole, University of Florence, 50019 Sesto Fiorentino, Italy; silvia.urciuoli@unifi.it; 5Institute of Agricultural Biology and Biotechnology, National Research Council (CNR), Via Moruzzi, 1, 56124 Pisa, Italy; 6Research Centre for Agriculture and Environment, Council for Agricultural Research and Economics (CREA), Via di Lanciola 12/A, 50125 Florence, Italy; 7Department for Sustainable Food Process, Università Cattolica del Sacro Cuore, Via Emilia Parmense 84, 29122 Piacenza, Italy

**Keywords:** colorectal cancer, PIRC rats, plasma biomarkers, secoiridoids, untargeted metabolomics

## Abstract

Oleuropein (OLE), a phenolic compound particularly abundant in the olive leaves, has been reported to have beneficial activities against colorectal cancer (CRC). In vitro studies suggested that these latter could be due to a modulation of the intestinal microbiota. Aiming to evaluate if OLE could affect the intestinal microbiota and the plasma metabolome, an antioxidant oleuropein-rich leaf extract (ORLE) was administered for one week to PIRC rats (F344/NTac-*Apc*^am1137^), a genetic model mimicking CRC. ORLE treatment significantly modulated the gut microbiota composition. Plasma metabolomic profiles revealed a significant predictive ability for amino acids, medium-chain fatty acids, and aldehydes. Pathway analysis revealed a significant decrease in phosphatidylcholine accumulation (LogFC = −1.67) in PIRC rats. These results suggest a significant effect of ORLE administration on faecal microbiota profiles and plasma metabolomes, thereby offering new omics-based insights into its protective role in CRC progression.

## 1. Introduction

In recent decades, there has been a burgeoning interest in exploiting the bioactive compounds present in olive leaves, particularly polyphenols, for their potential applications in the fields of nutraceuticals and biomedicine [1,2]. Specifically, oleuropein (OLE) represents the primary phenolic compound in olive leaves, together with other secoiridoids derived from the tyrosol structure, as well as flavonoids, lignans, and phenolic acids [3]. The phenolic composition of olive leaves is highly dependent on several terroir-associated variables, including geographical origin, type of cultivar, and tree age [4,5], while the phenolics recovered from olive leaf extracts can be strongly affected by the extraction method under consideration. In this scenario and following the health claim provided by EFSA in 2011, olive leaf extracts are now recognised as a popular nutraceutical taken as liquid or capsules [6]. Oleuropein (OLE), a secoiridoid phenolic compound abundant in olive leaves, has garnered considerable attention for its putative protective effects against colorectal cancer (CRC), the third most common cancer worldwide [7]. Besides the genetic component accounting for 35% of the total CRC cases [8], the incidence of the pathology is strongly affected by dietary habits (https://wcrf.org/diet-activity-and-cancer/, accessed on 10 June 2024), with consumption of specific foods linked to an increase or decrease in CRC risk, an effect that has also been explained by a modulation of the intestinal microbiome by the diet [9]. Regarding polyphenols, preclinical studies have suggested, in fact, that their beneficial activities may be mediated, in part, through modulation of the intestinal microbiota [10]. However, the precise mechanisms underlying this interaction and the subsequent impact on the host metabolome remain elusive. Once polyphenols are transported to the colon, they can be highly processed by the colonic microflora to produce aromatic acids [3]. After undergoing transformation in the small intestine and colon, phenolic metabolites enter the bloodstream and can subsequently reach the liver, where they undergo further modification [11]. Other phenolic metabolites are excreted in the form of bile components, being regenerated by intestinal microbial enzymes before re-absorption, while other unabsorbed metabolites are finally excreted in the faeces [10].

In our previous work [3], the oleuropein-rich leaf extract (ORLE) exhibited unique phenolic profiles and influenced the modulation of the faecal microbiota during simulated in vitro large intestinal fermentation. In vitro fermentation of ORLE led to an increase in hydroxytyrosol and other phenolic metabolites, significantly influencing the amino acid and fatty acid profiles of the faecal material. However, the ORLE impact on the composition of the intestinal microbiota in vivo remains unknown. Previously, we demonstrated that administering ORLE for one week to PIRC rats (F344/NTac-*Apc*^am1137^), an *Apc* mutated model mimicking CRC, resulted in a significant increase in tumour apoptosis and a down-regulation of proliferation, associated with the inhibition of nitric oxide (NO) and relative pro-inflammatory mediators expressed by tumour cells and inflammatory cells in the tumour microenvironment [12]. This finding suggests the possibility of testing ORLE as a complementary therapy in combination with standard anti-cancer drugs. Given these background conditions, the present study aims to elucidate the effects of an antioxidant oleuropein-rich leaf extract (ORLE) on the gut microbiota composition and plasma metabolomic profiles in a rat model that mimics CRC (PIRC rats carrying a heterozygous germline mutation in the *Apc*, the key gene in CRC development). The PIRC rat spontaneously develops multiple tumours in the colon and small intestine, thus standing as a robust model to study the protective effect of ORLE, derived from olive leaves, on CRC progression [10]. Therefore, by employing a multi-omics approach, integrating 16S rRNA gene sequencing and untargeted metabolomics, this research endeavours to provide novel insights into the protective role of ORLE against CRC progression. The findings may inform future therapeutic strategies and contribute to the development of targeted interventions leveraging the synergistic interplay between dietary antioxidants, gut microbiota, and host metabolism.

## 2. Materials and Methods

### 2.1. Preparation of the Olive Leaf Extract and Antioxidant Activity

The ORLE powder, derived from organic olive (i.e., Frantoio and Leccino cultivars) leaves, was obtained through an extraction technology based on membrane purification and vacuum concentration steps, as described in Romani et al. [13]. The characterisation of the phenolic profile of ORLE was previously done by HPLC-DAD-MS (high-performance liquid chromatography coupled with diode-array detection and mass spectrometry), revealing a total phenolic content of the dry extract equal to about 400 mg/g, of which oleuropein was the main compound (about 298.5 mg/g) [12]. The antioxidant activity of ORLE was determined using the DPPH (1,1-diphenyl-2-picrylhydrazyl) assay as previously described by [14,15], with some modifications. The antioxidant activity was expressed as Efficient Concentration (EC50 = mg of ORLE/mg DPPH), i.e., the concentration of ORLE necessary to reduce the initial concentration of DPPH by 50%.

### 2.2. Animal Experimental Design and Diets

PIRC rats (F344/NTac-*Apc*^am1137^) from the National Institutes of Health (NIH), Rat Resource and Research Center (RRRC) (University of Missouri, Columbia, MO, USA), and wild-type rats (WT Fisher F344) were bred in Ce.S.A.L. (Housing Centre for Experimental Animals of the University of Florence, Florence, Italy) in accordance with the Commission for Animal Experimentation of the Italian Ministry of Health (EU Directive 2010/63/EU for animal experiments), as described in Ruzzolini et al. [12]. Rats were maintained in polyethylene cages and fed with a standard AIN-76 diet (Laboratorio Dottori Piccioni, s.r.l., Gessate, MI, Italy). Eleven PIRC rats aged 12 months were randomly assigned to a control AIN-76 diet (control group, namely PIRC-CTR), while 10 PIRC rats (same age) were randomly assigned to the ORLE group (PIRC-ORLE), fed the same AIN-76 diet containing 2.7 g of ORLE/kg of diet. Additionally, 10 wild-type rats aged 12 months were fed the AIN-76 diet (WT-CTR), and 9 wild-type animals were assigned to the ORLE group (WT-ORLE). Considering that rats eat about 11 g of diet/day (mean body weight = 300 g), we administered a dose of ORLE of about 100 mg/kg b.w. This dose in rats corresponds to a human equivalent dose (HED) of 16 mg/kg [16]. To evaluate the changes in the intestinal microbiota, we collected faecal pellets at different treatment times (4 and 7 days), while for the plasma metabolite composition, the experiment was carried out only in the following experimental groups: PIRC-CTR (7 animals), PIRC-ORLE (6 animals), and WT-CTR (6 animals). In this case, after one week of treatment, rats were euthanised by CO_2_ asphyxia, and the blood was quickly collected by decapitation in a tube pre-treated with a 3.8% Na-citrate solution. After blood collection, the plasma samples were obtained by centrifugation (10 min, 680 rcf at 4 °C) and stored at −80 °C until further analyses. For the faecal microbiota analysis, the faecal pellets were collected in RNAlater (Qiagen, Hilden, Germany) and stored at −80 °C until DNA extraction. More details regarding animal maintenance and related information are available elsewhere [12].

### 2.3. Bacterial DNA Extraction, 16S rRNA Gene Sequencing, and Sequencing Data Analysis

The extraction of bacterial DNA from faecal material, together with the sequencing and corresponding data analysis, were carried out as reported in detail by Vitali et al. [17]. Briefly, DNA extraction was done using the DNeasy PowerLyzer PowerSoil Kit (Qiagen, Hilden, Germany). An Illumina MiSeq platform with a 300-bp paired-end reads protocol was used to perform the sequence of the hypervariable region V3–V4 of the 16S rRNA gene [14]. The pre-processing of the obtained reads was done using CUTADAPT (https://cutadapt.readthedocs.io/en/stable/, accessed on 12 January 2024), while SICKLE (https://github.com/najoshi/sickle, accessed on 12 January 2024) was used to remove reads showing low-quality portions [17]. Finally, OTUs/ASVs identification was done in MICCA (ver. 1.7.2), according to the UNOISE3 algorithm, while taxonomy was assigned by means of the RDP classifier (version 2.11) against the RDP database [17].

### 2.4. Faecal Water Preparation and Quantification of Short- and Medium-Chain Fatty Acids

The collected faecal material was stored at −80 °C. Thereafter, faecal short- and medium-chain fatty acids (SCFAs and MCFAs, respectively) were determined using a GC-MS method, based on the Agilent GC-MS system composed of a 5971 single quadrupole mass spectrometer, a 5890-gas chromatograph, and a 7673 autosampler as previously described [17].

### 2.5. Extraction of Metabolites from Plasma Samples

The extraction of plasma metabolites was done according to Saigusa et al. [18], with some modifications. Briefly, each thawed plasma sample (100 µL aliquot) was extracted with 400 µL of an acetonitrile/methanol solution (1/1 *v*/*v*) using an ultrasound-assisted extraction system (UAE; DU-32 ARGOLab, Milan, Italy; maximum power 120 W) for 10 min, considering three replicates (n = 3). After sonication, the plasma extracts were centrifuged in 1.5 mL tubes at 12,000× *g* for 15 min at 4 °C to remove large biomolecules (such as proteins). The supernatants were filtered through a 0.20 μm cellulose syringe filter in amber vials until further instrumental analysis. Also, a small aliquot of each plasma extract (20 µL) was taken and combined in the same UHPLC vial to provide a pooled quality control (QC) sample, required for the annotation-based approach and to check the instrumental variability.

### 2.6. Untargeted Metabolomic Profiling by High-Resolution Mass Spectrometry

The untargeted UHPLC-HRMS analysis was done using a Q Exactive Focus Hybrid Quadrupole-Orbitrap Mass Spectrometer (Thermo Scientific, Waltham, MA, USA) coupled to a Vanquish UHPLC system (Thermo Scientific) [19]. All the accurate details regarding the chromatographic separation, full scan MS analysis of the plasma extracts, and data-dependent (Top N = 3) MS/MS analysis of pooled quality control samples, can be found in Rocchetti et al. [19]. The MS-DIAL software (version 4.90) was then used to process the raw instrumental data, using an automatic peak finding, LOWESS normalisation, and annotation via spectral matching against the Mass Bank of North America (MoNA) and Phenol-Explorer databases (considering both parent compounds and the intestinal metabolites). The 80–1200 *m*/*z* mass range was searched for features with a minimum peak height of 10,000 counts per second, while the MS and MS/MS tolerances for peak centroiding were 0.01 and 0.05 Da, respectively. The identification step was based on mass accuracy, isotopic pattern, and spectral matching (i.e., following a level 2 of confidence in the annotation). Finally, the peak finder algorithm was used to fill in missing peaks, considering a 5-ppm tolerance for *m*/*z* values.

### 2.7. Statistical Analyses of Metabolomics and Metagenomics Data

Once identified, the compounds resulting from metabolomics were elaborated through multivariate statistics, considering 3 different software, namely Mass Profiler Professional (version: B.12.06; Agilent Technologies, Santa Clara, CA, USA), MetaboAnalyst 6.0, and SIMCA 13 (Umetrics, Umea, Sweden), as previously reported [19]. The metabolomics dataset was Log_10_-normalised, Pareto-scaled, and then elaborated using unsupervised and supervised multivariate statistics, namely on one side the unsupervised hierarchical cluster and principal component analyses, and on the other side the supervised orthogonal projections to latent structures discriminant analysis (OPLS-DA), respectively. The OPLS-DA model was also checked for validation parameters (considering a prediction ability > 0.5), inspected for outliers, and excluded for overfitting [19]. The prediction ability of each plasma metabolite was recorded through a variable importance in projection (VIP) method, using as the minimum significant threshold a score > 0.8; furthermore, volcano plot analyses were done by coupling FC analysis (cut-off value > 1.2) and ANOVA (*p*-value < 0.05) on selected pairwise comparisons.

As far as the statistical analysis of the metagenomic data is concerned, it was performed using R software (https://www.r-project.org/, accessed on 12 January 2024). Accurate details regarding the elaboration of sequencing count data are available in Vitali et al. [17]. Also, the alpha and beta diversity in bacteria were evaluated using three indices, namely species richness, evenness index, and Shannon index [17]. The beta diversity of bacterial communities was plotted using Principal Coordinate Analysis (PCoA) ordinations according to the Bray–Curtis dissimilarity [17]. The association of bacterial community diversity with experimental variables under investigation was evaluated through a permutational analysis of variance (PERMANOVA: 9999 permutations, Bray–Curtis dissimilarity), while the LEfSe (linear discriminant analysis effect size) analysis was used to identify potential bacterial biomarkers separating the different group treatments [17].

### 2.8. Pathway Analysis and Plasma Metabolites as Biomarkers

To search for plasma biomarker compounds of CRC, the online tool MetaboAnalyst 6.0 was used to inspect how the most important pathways represented by the metabolites annotated (using the pathway library: *Rattus norvegicus*, Kyoto Encyclopaedia of Genes and Genomes, KEGG) were affected by ORLE treatment. Additionally, the pathway analysis was used to provide a metabolite set enrichment based on the discriminant and significant class of metabolites/metabolic pathways outlined by the multivariate statistics.

## 3. Results and Discussion

### 3.1. Intestinal Microbiota Profiles

We conducted species richness and biodiversity analyses to assess variations in bacterial communities among faecal samples from different experimental groups. Additionally, to investigate the potential impact of treatment duration on microbiota profiles, faecal samples were collected from all experimental groups at both 4 and 7 days of treatment. The analysis, presented in Appendix A, revealed significant differences in richness among the experimental groups in both PIRC and WT rats (ANOVA, Appendix A), while significant differences in biodiversity assessment (Simpson’s dominance index) were evident for WT animals only (ANOVA, Appendix A). In detail, an increase in richness in 4-day-treated WT animals with ORLE was depicted, while an increase in bacterial biodiversity (Simpson’s dominance index) was observed after 7 days in WT animals in both conditions (CTR and ORLE). Furthermore, upon examining the evenness and Shannon diversity (Appendix A), no significant differences were observed among the experimental groups.

To assess the variability of microbial communities among samples (i.e., beta diversity), we conducted Principal Coordinates Analysis (PCoA) based on Bray–Curtis distances (Figure 1).

The PCoA ordination revealed that the distribution of samples from both PIRC and WT rats was influenced by the duration of the experimental treatment with ORLE, with samples collected after 4 days distinctly separated from those collected after 7 days. This observation was further supported by a PERMANOVA analysis (Table 1), which showed a significant association with the length of treatment (“length” variable; *p*-value = 0.001). Importantly, on the second ordination axis (Figure 1), discrete sample clusters were evident following ORLE administration, as also highlighted by PERMANOVA analysis (Table 1, *p*-value = 0.002). Moreover, as previously reported [14], a difference attributed to the genotype was also present (Table 1), even though the value did not reach statistical significance.

In the taxonomic structure of the bacterial community (Figure 2A), Firmicutes dominated in WT rats (42.4% in CTR and 46.58% in ORLE), while Bacteroidetes prevailed in PIRC rats (30% in CTR and 32.072% in ORLE). Bacteroidetes phylum was the second most abundant in WT (24.5% in CTR and 23.76% in ORLE) and Firmicutes in PIRC (32.7% in CTR and 28.52% in ORLE). Actinobacteria showed similar abundance across samples (0.36% in WT CTR and 1.8% in WT ORLE; 0.45% in PIRC CTR and 1.0% in PIRC ORLE). Proteobacteria were more abundant in PIRC (8.24% in CTR and 12.5% in ORLE) than in WT (4.01% in CTR and 5.09% in ORLE). Verrucomicrobia were well represented in all groups (21.8% to 29.84% in PIRC, 21.83% to 26.8% in WT).

A linear discriminant analysis effect size (LEfSe) analysis was then conducted to identify the key bacterial phylotypes that were differentially represented among the two groups of rats treated with ORLE (both genotypes) compared to control rats not treated (CTR) (Figure 2B). Specifically, at the class level, Clostridiales were enriched in the faecal bacterial community of ORLE-treated rats. It was found that after 7 days of ORLE treatment, at the genus level, an enrichment of Paraprevotella, Anaerotruncus, Oscillibacter, and Sporobacter was present. Regarding Paraprevotella, data available in the literature [20] document that this genus is increased in rats treated with *Portulaca oleracea* polysaccharides, in which a reduction in harmful bacteria (Escherichia/Shigella and Bacteroides) was also observed, thus suggesting (in agreement with our results) that the enrichment of Paraprevotella in ORLE-treated rats may have a beneficial effect. Regarding Anaerotruncus, they are well-known butyrate-producing bacteria, and it has been demonstrated [21] that they produce various hepatoprotective compounds (including biotin, ornithine, arginine, spermidine, isoleucine, and valine) in mice with damaged livers. The presence of Oscillibacter in the faecal microbiota, as observed in our results, has been linked to the metabolism of dietary components, particularly carbohydrates and fibres. It may play a role in the fermentation of these substrates, contributing to the production of SCFAs and other metabolites that can influence gut health and host metabolism. In particular, Oscillibacter has been shown to have anti-inflammatory functions and play crucial roles in the maintenance of mucosal homeostasis in humans and mice [22,23]. Besides, Alistipes (from the Rikenellaceae family) and Morganella were more abundant in the faecal bacterial community of the CTR group. Finally, at the faecal level, despite the presence of genera producing SCFAs, the analysis conducted through GC-MS for faecal SCFA and MCFA composition was not affected by ORLE treatment.

### 3.2. Metabolomic Profile and Multivariate Discrimination of Plasma Samples

The analysis of the microbiota composition, including beta diversity and relative abundances, indicated that ORLE treatment significantly affected the intestinal microbiome, while only slight differences were observed between PIRC and WT rats. Given that we previously observed the beneficial effect of ORLE in PIRC rats [12], in this study plasma metabolomics was exploited to exclusively highlight the impact of ORLE in PIRC rats, for which no prior data are available, thus serving as the primary focus of our investigation. Therefore, we focused the analysis of plasma composition solely on the PIRC animals, comparing them to the WT rats fed a CTR diet. Before running the metabolomics analysis, the in vitro antioxidant activity of the ORLE extract administered to PIRC rats was first investigated. The results of the DPPH assay revealed that the EC50 was equal to 1.03 mg ORLE/mg DPPH. However, as reported in the scientific literature, several factors, such as olive cultivar, geographic location, leaf pre-treatment, extraction method, total solid content of the extract, and expression of the results, can affect the antioxidant activity of OLE, thus hampering the comparison between different studies [24].

The untargeted metabolomics profile of plasma samples was assessed using a UHPLC-HRMS approach, leading to the annotation of 464 metabolites according to a level 2 of confidence (COSMOS coordination of Standards in Metabolomics). The annotation step was performed by exploiting two comprehensive databases: the Mass Bank of North America (MoNA) and Phenol-Explorer. The latter was particularly useful for identifying potential intestinal and bioavailable metabolites of phenolic compounds provided by the ORLE-supplemented diet. A detailed list of all the annotated metabolites is available in the Appendix A. Overall, the chemical classes explained by the metabolomics dataset were determined through chemical similarity enrichment, showing a significant abundance of amino acids, diacylglycerophosphocholines, flavonoid glycosides, and other phenolic subclasses (such as hydroxycinnamics, flavanones, and curcuminoids). Of interest, the annotation strategy facilitated the identification of OLE, oleuropein aglycone (OLE-Aglycone), and other colonic metabolites of parent phenolic compounds, such as lower molecular weight phenolics classified as phenolic acids (hydroxycinnamic and hydroxybenzoic derivatives) and other metabolites (including hippuric acid).

Considering the huge amount of potential information associated with the annotated mass features, a multivariate statistical approach was used to cluster plasma samples and derive biological meaning. As the initial step, we employed the sPLS-DA algorithm to mitigate model sparseness by reducing the number of significant variables (metabolites), thereby generating a more robust and interpretable score plot. As depicted in Figure 3A, the two principal components (PC1 and PC2) were able to cumulatively explain 19.2% of the total variability, providing some degree of overlap among the three sample groups under investigation. To mitigate sample variability and isolate the specific impact of ORLE on the PIRC metabolome, we exploited a multivariate statistical approach based on the supervised OPLS-DA. This method effectively separates variation not directly correlated with Y in the X matrix (i.e., orthogonal signal correction), considering solely Y-predictive variation and thereby maximising the covariance between groups. Three sample groups were considered: Group 1 (PIRC-ORLE), Group 2 (PIRC-CTR), and Group 3 (WT-CTR). Indeed, the OPLS-DA score plot (Figure 3B) distinctly discriminated between dietary conditions across the different experimental groups. In particular, the score plot showed the PIRC-ORLE group on the right side of the orthogonal latent vector, while WT rats clustered on the left side, effectively discriminating the variability within the PIRC-CTR and WT groups. Additionally, the prediction model built was characterised by more than acceptable goodness parameters related to goodness-of-fit (R^2^X = 0.509 and R^2^Y = 0.981) and cumulative goodness of prediction (Q^2^ = 0.790). Besides, the model was cross-validated by using a cross-validation ANOVA (*p*-value = 3.4 × 10^−22^) and excluded for both significant outliers (according to Hotelling’s T^2^ distribution) and overfitting (using permutation testing based on 100 random permutations) (Appendix A). As a general consideration, the OPLS-DA score plot outlined a potential effect of ORLE in modulating the plasma metabolomic profile, considering the between-variability separation explained by the prediction model.

Therefore, the most discriminant plasma metabolites outlining the separation observed (Figure 3B) were extrapolated through a VIP selection method approach. Overall, a total of 216 plasma metabolites were characterised by a VIP score > 1 (i.e., extremely discriminant for prediction purposes), with 2-methylbutyrylglycine and pipecolic acid showing the highest discriminant potential, being characterised by VIP scores of 1.68 and 1.66, respectively. The compound 2-methylbutyrylglycine is an acyl glycine, normally representing a minor metabolite of fatty acids. According to the Human Metabolome Database (HMDB) [25], the excretion of certain acyl glycine in biofluids can be indicative of disorders related to mitochondrial fatty acid beta-oxidation. Regarding the role of pipecolic acid as a discriminant biomarker, it represents a metabolite of lysine. However, it remains uncertain whether it originates directly from food intake or from mammalian or intestinal bacterial enzyme metabolism [25]. Going into detail, plasma pipecolic acid, particularly the D-isomer, is reported to originate mainly from the catabolism of dietary lysine by intestinal bacteria rather than by direct food intake. Interestingly, among the first 50 discriminant VIP metabolites (Appendix A), we listed five amino acids, namely histidine (VIP score = 1.31), tryptophan (VIP score = 1.41), threonine (VIP score = 1.45), arginine (VIP score = 1.50), and lysine (VIP score = 1.59), and several polyphenols (mainly phenolic acids and flavonoids) and their metabolites (such as caffeic acid and its 3-*O*-glucuronide, ferulic acid and its 4-O-glucuronide, and others).

Of interest, Ole-aglycone was recorded among the discriminant compounds of this prediction model, characterised by a significant VIP score of 1.07 and a LogFC value of 2.4 for the comparison “PIRC-ORLE” vs. “PIRC-CTR”. Another interesting metabolite generally recognised as the biomarker of phenolics-rich diets (arising from the combination of benzoic acid and glycine) was hippuric acid [25]. Under our experimental condition, this metabolite was outlined as a biomarker of the PIRC-ORLE group, recording a LogFC of 0.9 when compared with the PIRC-CTR group. As previously reported [3], there is limited detailed information available on the correlation between the colonic pathway of OLE and its plasma bioavailability. As a matter of fact, hippuric acid can be produced through the colonic metabolisation of OLE and its aglycone, leading to the formation of hydroxytyrosol and, subsequently, hydroxyphenylacetics and hydroxybenzoics derivatives resulting from both dihydroxylation and oxidation mechanisms [26]. The substantial increase in plasma hippuric acid levels following one week of ORLE administration to PIRC rats appears to support the significant impact of this extract in terms of its potent anti-inflammatory activity in the colon, as previously highlighted [12]. Regarding other discriminant metabolites, we found an impact on polyamines (such as spermidine; VIP score = 1.49) and plasma fatty acids and lipid derivatives, with 13,16,19-docosatrienoic acid, stearoylcarnitine, and decanoic acid being highly discriminant (VIP score > 1.3).

Therefore, taken together, the findings derived from this preliminary OPLS-DA model illustrated the effectiveness of our metabolomics dataset in elucidating biochemical perturbations of amino acids, polyphenols, and fatty acid derivatives. This underscores the necessity for conducting comprehensive pathway analyses to uncover the biological significance of these observations. To extrapolate the discriminant markers of each possible comparison, namely “PIRC-ORLE vs. PIRC-CTR”, “PIRC-ORLE vs. WT group”, and “PIRC-CTR vs. WT group”, we carried out three additional OPLS-DA models that are reported in the Appendix A. Overall, each additional model provided excellent parameters related to the goodness of prediction, recording Q^2^(cum) values > 0.78 and outlining clear separation trends between the different experimental groups. Thereafter, the VIP discriminant metabolites of each prediction model were crossed through a Venn analysis to evaluate both common and exclusive compounds of each possible comparison. The results obtained are reported in Appendix A, showing 168 common discriminant metabolites (accounting for 41.7% of the total) when considering all constructed prediction models, with some metabolites exclusively representing specific comparisons (around 5% of the discriminant metabolites). Interestingly, among the metabolites exclusively identified in the “Group 1 vs. Group 2” comparison, several phenolic compounds passed the Volcano Plot analysis criteria (*p* < 0.05 and Fold-Change > 1.2). These include nepetin (a flavone), diosmin (a flavone), eriodictyol (a flavanone), lariciresinol-sesquilignan (a lignan), and sitosterol ferulate (belonging to hydroxycinnamic acids), thus confirming the hypothesis that ORLE added to the PIRC diet left a potential antioxidant signature on the plasma metabolome (Appendix A).

### 3.3. Pathway Analyses to Highlight the Impact of ORLE on Plasma Metabolome

The discriminant metabolites for the comparison “PIRC-ORLE vs. PIRC-CTR” were then loaded into the pathway analysis tool of MetaboAnalyst 6.0, and the changes in plasma metabolomic profile were evaluated against the metabolome of *R. norvegicus*. Overall, the most impactful and significant pathways belonged to glycerophospholipid metabolism (*p*-value < 0.01), followed by pyrimidine metabolism (*p*-value < 0.01), lysine degradation (*p*-value < 0.05), aminoacyl-tRNA biosynthesis (*p*-value < 0.05), phenylalanine metabolism (*p*-value < 0.05), alpha-linolenic acid metabolism (*p*-value < 0.05), and arginine biosynthesis (*p*-value < 0.05). The significant (*p*-value < 0.05) discriminant classes in plasma samples considering the different LogFC (fold-change) values for the comparison “PIRC-ORLE” vs. “PIRC-CTR” are reported in Table 2.

Overall, the chemical class represented by polyamines was significantly up-accumulated in the PIRC-ORLE group, with spermidine being strongly up-accumulated and discriminant in plasma samples, showing a VIP score > 1.8 and a LogFC value > 2 (Appendix A). A dysregulation of polyamine metabolism is associated with the development of CRC, although the underlying mechanisms are not fully understood. It has been reported that following CRC, the activities of polyamine-synthesising enzymes and polyamine content increase 3–4-fold compared to levels found in equivalent normal colonic mucosa. Consequently, polyamines have been recognised as markers of neoplastic proliferation [27].

Regarding other discriminant classes, our findings indicated a significant decrease in amino acid accumulation (Table 2) for the comparison between PIRC-ORLE and PIRC-CTR, with tryptophan being the most discriminant compound (LogFC = −0.67; Appendix A). Also, lysine showed the most significant variation (*p*-value < 0.01). On the other hand, a significant increase in phenolic metabolites was observed, with glucuronide of dihydroferulic acid being highlighted as the most discriminant biomarker in the PIRC-ORLE group (LogFC = 1.25; Appendix A). Finally, thymine, 3-hexyl-pyridine, terpinyl-isovalerate, and ribose 5-phosphate emerged as the most discriminant metabolites of their corresponding chemical classes (Table 2).

As far as the distribution of glycerophospholipids is concerned, the overall level of phosphatidylcholine (PC) is reported to increase in CRC [28]. Previous studies have determined that CRC alters the phospholipid composition of the cell membrane [29]. These alterations can influence cell proliferation, viability, and tumour development. While PC is the most dominant phospholipid in both non-neoplastic and cancer tissues, its abundance is notably increased in colorectal cancer cells [28]. Interestingly, the pathway analysis showed that plasma samples from PIRC-ORLE exhibited a significantly (*p*-value < 0.05) lower accumulation of PC (LogFC = −1.67; Appendix A) compared to PIRC-CTR, indicating a pronounced effect of ORLE administration on this biomarker in PIRC rats. Also, glycerophospholipids showed an overall down-accumulation (Table 2) for the selected comparison, with LysoPC(16:0) outlined as the most discriminant metabolite (VIP score = 1.83). This effect could be due to the inhibitory activity mediated by the antioxidant properties of ORLE and colonic metabolites on key enzymes involved in phospholipid biosynthesis, such as phospholipase A2 (PLA2) and lyso-PC acyltransferase (LPCAT) [30]. However, further ad hoc studies are required to confirm this hypothesis. Interestingly, we found an opposite trend for fatty acids and conjugates, indicating an overall increase in the PIRC-ORLE group (Table 2).

In recent years, the application of metabolomics in cancer research has led to a renewed understanding of metabolism’s role in cancer development and progression, enabling researchers to identify novel cancer-causing metabolites and biomarkers [31]. Geijsen et al. [32] previously reported findings from an international consensus on plasma metabolites associated with different CRC stages. These authors identified sphingolipid-derivatives, phosphatidylcholine- and lysophosphatidylcholines-derivatives, citrulline, and histidine as key plasma metabolites implicated not only in CRC development but also in its progression. In our experimental setting, we observed a significant discriminant ability not only for PC (*p*-value < 0.01; Appendix A), but also for histidine (*p*-value < 0.01; Appendix A). Higher levels of histidine were detected when comparing the PIRC-ORLE and PIRC-CTR groups vs. the WT group, recording LogFC values of 0.81 and 0.78, respectively. Histidine is associated with aspartate metabolism and is one of the amino acids involved in the tricarboxylic acid cycle. The tricarboxylic cycle has been reported in CRC development, with differences observed between colorectal tumour tissue and normal mucosa. The scientific literature has shown that histidine concentration tends to be lower in advanced CRC patients [32]; however, systemic inflammation plays a significant role in determining the final histidine concentration. The histidine up-accumulation values detected in this work are difficult to realistically compare with the existing scientific literature, considering the conflicting results about histidine as a valid CRC plasma biomarker [32].

As the next step, we carried out a pathway analysis for the comparison “PIRC-CTR vs. WT group” to better evaluate the impact of ORLE on the plasma metabolomic profile of rats (Appendix A). The most impactful and significant pathways belonged to sphingolipid metabolism (*p*-value < 0.01), phenylalanine metabolism (*p*-value < 0.01), aminoacyl-tRNA biosynthesis (*p*-value < 0.01), glycerophospholipid metabolism (*p*-value < 0.01), lysine degradation (*p*-value < 0.05), alpha-linolenic acid metabolism (*p*-value < 0.05), and arginine biosynthesis (*p*-value < 0.05). Regarding PC accumulation, coherently with the trend previously reported for the impact of ORLE on PC reduction, the pathway analysis showed that plasma samples belonging to PIRC-CTR have a significantly (*p*-value < 0.05) higher accumulation (LogFC = 0.32; Appendix A) of PC when compared with the WT group, thus confirming the positive effect exerted by ORLE on this CRC biomarker.

### 3.4. Importance of Lipid-like Molecules as Related to the ORLE Effect on Plasma Metabolome

Looking at the results of volcano plot analysis and pathway analyses carried out against the metabolome of *R. norvegicus* (Appendix A), it was clear that lipid-like molecules (such as glycerophospholipids and fatty acids) emerged as the most impactful classes at the biochemical level. In particular, the class of phospholipids showed an average down-accumulation trend in each selected comparison and was identified as a marker of the PIRC group. The most significant phospholipids were represented by several subclasses, namely unsaturated phosphatidylethanolamines, phosphatidylserines, such as PS (20:0/22:0), and phosphatidylcholines, including PC(20:1(13Z)/20:1(13Z)) and PC(18:1(11Z)/20:1(11Z)) (Appendix A).

Other lipid biomarkers associated with PIRC-ORLE were represented by docosatrienoic acid (VIP score = 1.49) and 3-hydroxy-1-phenyl-1-heneicosanone (VIP score = 1.13). Furthermore, lipid peroxidation products (such as the aldehydes 2,4-decadienal and 3,6-undecadienal) also showed significant variations (Table 2). Lipid peroxidation and subsequent formation of toxic aldehydes, such as 4-hydroxynonenal and MDA, are known to be involved in numerous pathophysiological processes, possibly including the risk and development of CRC [33]. Under our experimental setting, we found a significant variation of 2,4-decadienal (Appendix A), an aldehyde directly deriving from the peroxidation of alpha-linoleic acid [34]; however, by evaluating the comparison “PIRC-ORLE vs. PIRC-CTR”, it seems that the phenolic-rich extract was not effective in reducing the accumulation of this compound, thus suggesting higher oxidation of the ω6 alpha-linoleic acid. Regarding the ω3 alpha-linolenic acid, it was included by the volcano plot analysis as a significant biomarker for the different pairwise comparisons; however, no direct peroxidation product of this fatty acid was detected, making it difficult to speculate about possible biochemical perturbations. Overall, the detection and quantification of such species in different biofluids are of great interest. However, due to their volatility (especially for short-chain aldehydes) and their reactive properties (e.g., binding, decomposition), the analysis of aldehydes remains challenging, requiring targeted approaches [33].

Another plasma metabolite of great interest is cholesterol, a sterol lipid, which exhibited noteworthy variations according to the volcano plot analysis (Appendix A) and is outlined as a biomarker of PIRC rats. Extensive literature [35,36] has studied plasma cholesterol and cholesterol-related metabolic pathways in relation to CRC. However, ad hoc studies have revealed weak associations between plasma cholesterol levels and CRC development.

Furthermore, as far as octanoic acids are concerned, plasma metabolomics revealed a good discriminant ability for 7-hydroxyoctanoic acid (Appendix A). As a general consideration, 7-hydroxyoctanoic acid is an (omega-1)-hydroxy-fatty acid that is octanoic acid in which a hydroxyl group has replaced the 7-pro-R hydrogen. In this work, 7-hydroxyoctanoic acid was significantly down-accumulated when considering the comparison “PIRC-ORLE vs. PIRC-CTR” (LogFC = −0.46; *p*-value < 0.05), thus indicating a potential impact of ORLE on the plasma distribution of this MCFA. Regarding decanoic acid, plasma metabolomics revealed the presence of high levels of decanoic acid in samples belonging to the PIRC-ORLE group (LogFC = 1.03; *p* < 0.05) when compared with the other dietary groups. The presence of these metabolites, combined with their statistical significance, is consistent with the rapid intestinal absorption of MCFAs and their transportation into the bloodstream via the portal vein. Once in circulation, they are known to reach the liver and undergo metabolism through beta-oxidation in mitochondria [37]. Regarding decanoic acid, it has been proposed as a valuable plasma diagnostic biomarker of CRC [38]. We found that the addition of ORLE to the PIRC diet had no significant effect on counteracting the increase in this metabolite, likely due to the advanced stage of tumourigenesis in these PIRC rats. This is consistent with the overall up-accumulation of fatty acids and conjugates reported in Table 2.

### 3.5. Limitations of the Study

The findings of this study demonstrate a significant modulatory effect of ORLE administration on the gut microbiota composition and plasma metabolomic profiles in PIRC rats, a genetic model mimicking colorectal cancer. Notably, the treatment duration emerged as a critical factor influencing the microbial community structure, with distinct clustering observed between samples collected at different time points (4 days vs. 7 days). This observation underscores the dynamic nature of the gut microbiome and highlights the importance of considering temporal aspects in future investigations. While the present study provides valuable insights into the potential mechanisms underlying the protective effects of ORLE against CRC progression, it is imperative to acknowledge certain limitations.

The sample size, although adequate for the primary analyses, may restrict the statistical power and generalizability of the findings. Additionally, the lack of functional characterisation of the observed microbial shifts and their direct impact on host physiology warrants further exploration. Future studies should consider incorporating multi-omics approaches, such as metatranscriptomics and metabolomics of faecal samples, to elucidate the functional implications of the altered gut microbiome. Another potential limitation lies in the inability to discern the specific contributions of individual phenolic compounds present in the ORLE extract. While oleuropein is the predominant component, the synergistic or antagonistic effects of other phenolics cannot be ruled out. Targeted interventions with purified compounds or fractionated extracts may provide more refined insights into the structure-activity relationships governing the observed biological effects. Furthermore, the translational potential of these findings for human populations remains to be determined. Validation in clinical cohorts, accounting for inter-individual variability in gut microbiome composition and dietary patterns, is crucial for assessing the feasibility of ORLE supplementation as an adjuvant therapy in CRC management. Despite these limitations, the present study contributes to the growing body of evidence supporting the beneficial potential of olive leaf polyphenols against colorectal cancer. The observed modulation of the gut microbiome and host metabolome by ORLE treatment provides a mechanistic foundation for future investigations aimed at developing targeted interventions that leverage the intricate interplay between dietary antioxidants, gut microbial communities, and host metabolism in the context of chronic diseases such as cancer.

## 4. Conclusions

The current study aimed to investigate the impact of a one-week ORLE-enriched diet on the intestinal microbiota and plasma metabolomic profiles of PIRC rats. In terms of faecal microbiota composition, ORLE administration shapes the bacterial communities in both PIRC and WT rats, with a significant increase in Clostridiales in the ORLE-treated group. Interestingly, variations in plasma metabolomes were observed, with specific LogFC values associated with the ORLE-based diet, indicating potential modulation of cancer-related compounds. The most discriminant biomarker compounds identified upon ORLE administration to PIRC rats mainly belonged to phospholipids, amino acids, phenolic metabolites, polyamines, and medium-chain aldehydes/fatty acids. Furthermore, the ORLE-based diet predominantly influenced the plasma metabolome profile of PIRC rats compared to the CTR group, offering new insights into the protective role of oleuropein and its colonic metabolites against CRC progression. Future dedicated studies on these identified biomarker compounds hold promise for a deeper understanding of the underlying biochemical mechanisms involved.

## Figures and Tables

**Figure 1 antioxidants-13-00724-f001:**
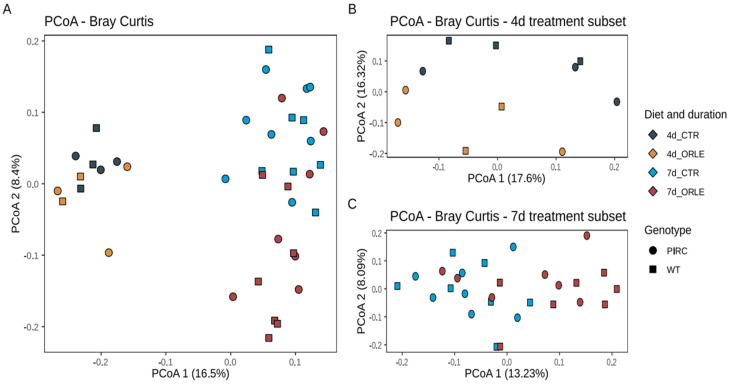
PCoA ordinations based on Bray–Curtis dissimilarity index. (**A**) Ordination of all samples after 4 and 7 days of ORLE treatment in PIRC and WT genotypes. (**B**,**C**) PCoA ordinations based on the Bray–Curtis dissimilarity index considering subsets of 4 and 7 days of ORLE treatment, respectively. The shape of the points indicates the sample genotype: squares for WT, circles for PIRC. The colour of the points represents different experimental conditions considering the treatment (CTR or ORLE) and the duration (4 or 7 days).

**Figure 2 antioxidants-13-00724-f002:**
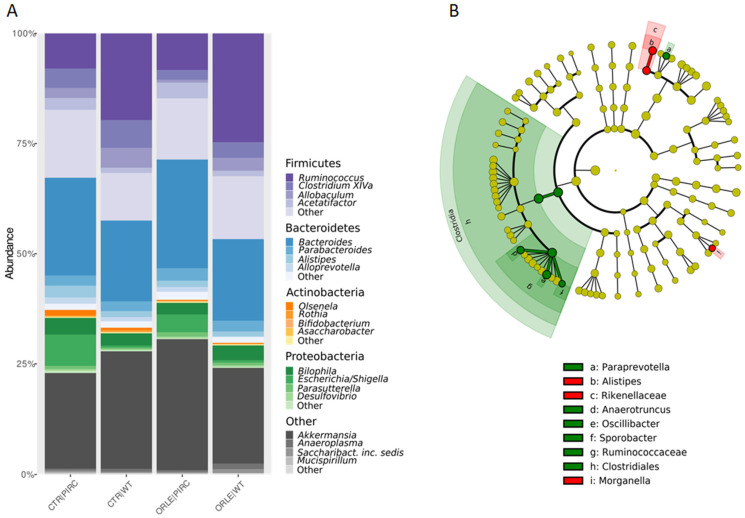
(**A**) Relative abundances of phyla and genera (only phyla and genera with abundance > 5% in at least one sample were represented). Composition of bacterial communities at the phylum level, grouped for sample type (CTR vs. ORLE) and genotype (PIRC vs. WT). (**B**) LEfSe on faecal bacterial communities. The cladogram shows the most discriminative bacterial clades after ORLE treatment. The coloured regions/branches indicate differences in the bacterial population structure between the different groups (CTR in red and ORLE in green). Statistically significant taxa enrichment among groups was obtained with Kruskal–Wallis test among classes (alpha = 0.05). The threshold for the logarithmic LDA score was 2.0.

**Figure 3 antioxidants-13-00724-f003:**
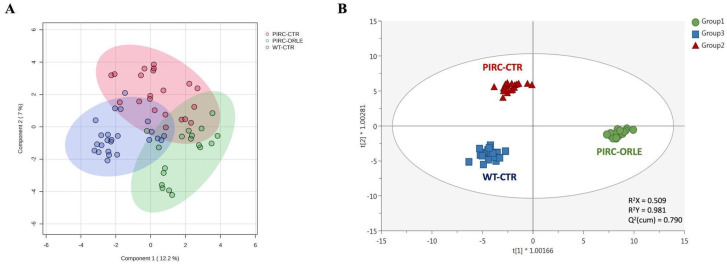
sPLS-DA (**A**) and OPLS-DA (**B**) score plots showing the modifications of the metabolomic profile of plasma samples belonging to Group 1 (PIRC-ORLE group, in green), Group 2 (PIRC-CTR group, in red), and Group 3 (WT-CTR group, in blue). t1 = latent vector 1; t2 = latent vector 2.

**Table 1 antioxidants-13-00724-t001:** Results of PERMANOVA analysis to test the effects of different sources of variation (ORLE treatment, genotype, length of treatment) on the Bray–Curtis distance matrix between all pairs (999 permutations, Bray–Curtis distance). * *p*-value < 0.05.

**Variables**	**Df**	**Sum of SQS**	**R^2^**	**F-test**	** *p* ** **-value**
ORLE treatment	1	0.2461	0.05346	2.5224	0.002 *
Genotype	1	0.1273	0.02767	1.3054	0.105
Length	1	0.7176	0.15589	7.3556	0.001 *
Residual	36	3.5119	0.76298		
Total	39	4.6029	1.00000		
**4-day Intervention Subset**					
**Variables**	**Df**	**Sum of SQS**	**R^2^**	**F-test**	** *p* ** **-value**
ORLE treatment	1	0.14121	0.14525	1.5535	0.020 *
Genotype	1	0.08233	0.08468	0.9057	0.644
Interaction	1	0.11236	0.11588	1.2361	0.143
Residual	7	0.63627	0.65449		
Total	10	0.97216	1.00000		
**7-day Intervention Subset**					
**Variables**	**Df**	**Sum of SQS**	**R^2^**	**F-test**	** *p* ** **-value**
ORLE treatment	1	0.21495	0.07866	2.2510	0.001 *
Genotype	1	0.12942	0.04736	1.3553	0.047 *
Interaction	1	0.09656	0.03534	1.0113	0.457
Residual	24	2.29174	0.83864		
Total	27	2.73268	1.00000		

**Table 2 antioxidants-13-00724-t002:** Significant (*p*-value < 0.05) discriminant classes in plasma samples considering the different cumulative LogFC (fold-change) values for the comparison “PIRC-ORLE” vs. “PIRC-CTR”. The most significant VIP biomarker of each class is also provided along with its VIP score. Sig. = significance.

Chemical Class	Sig. (*p*-Value)	PIRC ORLE vs. CTR	Plasma Biomarker (OPLS-DA)
Polyamines	4.11 × 10^−3^	LogFC: 1.24	Spermidine (VIP score = 1.82)
Amino acids and peptides	6.48 × 10^−3^	LogFC: −5.00	Tryptophan (VIP score = 1.37)
Fatty acids and conjugates	1.60 × 10^−2^	LogFC: 2.36	13,16,19-Docosatrienoic acid (VIP score = 1.89)
Glycerophospholipids	5.54 × 10^−3^	LogFC: −0.51	LysoPC(16:0) (VIP score = 1.83)
Medium-chain aldehydes	1.99 × 10^−2^	LogFC: 1.99	3,6-Undecadienal (VIP score = 1.02)
Phenolic metabolites	1.44 × 10^−2^	LogFC: 3.72	Dihydroferulic acid, 4-*O*-glucuronide (VIP score = 1.72)
Pyridines and derivatives	1.89 × 10^−2^	LogFC: −1.93	3-Hexyl-pyridine (VIP score = 1.34)
Pyrimidines and derivatives	2.21 × 10^−2^	LogFC: 0.03	Thymine (VIP score = 1.41)
Terpenoids	9.40 × 10^−3^	LogFC: −1.01	Terpinyl-isovalerate (VIP score = 1.45)
Carbohydrates and conjugates	2.44 × 10^−2^	LogFC: 2.25	Ribose 5-phosphate (VIP score = 1.32)
Other metabolites	1.06 × 10^−2^	LogFC: 0.30	13′-Hydroxy-tocopherol (VIP score = 1.23)

## Data Availability

The raw data presented in this study are fully available in the article and Appendix A.

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
