# Peer review of "Changes in Faecal Microbiota Profile and Plasma Biomarkers following the Administration of an Antioxidant Oleuropein-Rich Leaf Extract in a Rat Model Mimicking Colorectal Cancer"

_antioxidants, 2024, doi:10.3390/antiox13060724_

Round 1
Reviewer 1 Report
Changes of fecal microbiota profile and plasma biomarkers following the administration of an antioxidant oleuropein-rich leaf extract in a rat model mimicking colorectal cancer, uses a rat model of APC driven CRC (also SB tumors) and examines the impact of short-term administration of oleuropein-rich leaf extract on the composition of the fecal microbiota and plasma metabolite composition. This paper is a follow up to a previous study, examining the impact of the extract on tumor apoptosis and iNOS. This follows in vitro studies, looking for synergy between chemotherapeutic agents and the extract in cell line work. The group has further looked at other phenolic rich extracts in this rat model. The work extends to earlier findings showing that the extract alters the fecal microbiota in a way that might favor beneficial bacteria. Plasma metabolites are also significantly altered. However, it is not clear what the next move would be by this group based on the findings.
The work suggests that short term consumption of the extract altered the fecal microbiota, notably suppressing E. coli/Shig. The authors say Bacteroides was suppressed as well. However, that is not evident in Fig. 2B. Alpha and Beta diversity indexes show marginal changes, more pronounced in wt rats. One of the odd features is the fairly consistent difference in meta-parameters of the microbiota (Fig. S1) between 4 days and 7 days, although not reaching significance. Is there any known reason for this. Did the rats have new cage mates or was the housing otherwise altered when the experiment began?
The plasma metabolite study is fairly exhaustive and exhausting to follow. However, since the goal is to establish that the extract does alter the composition, the case is established.
My major concern is the dosage of the extract. It appears that the dose was somewhat higher than that used by de Bock M, Derraik JG, Brennan CM, Biggs JB, Morgan PE, Hodgkinson SC, Hofman PL, Cutfield WS.(Olive (Olea europaea L.) leaf polyphenols improve insulin sensitivity in middle-aged overweight men: a randomized, placebo-controlled, crossover trial. PLoS One. 2013;8(3):e57622. doi: 10.1371/journal.pone.0057622. Epub 2013 Mar 13. PMID: 23516412; PMCID: PMC3596374.) in their human intervention study. How do you justify the dosage?
It is not stated what your next move might be to use the findings from this study. Are you contemplating some sort of fecal transplant studies, without ORLE intervention directly in the study animals-(transplanting from OLRE fed animals into untreated animals) to determine whether the alteration in the microbiota impact tumor growth?
Are you going to show any enhanced effect on suppressing tumorigenesis in the PIRC rats using OLRE and some other agents such as those used by Beach R, Hatano Y, Qiao Y, Grady J, Sei S, Mohammed A, Rosenberg DW. Combination of naproxen and a chemically-stable eicosapentaenoic acid analog provide additive tumor protection in Pirc rats. Int J Cancer. 2023 Jun 15;152(12):2567-2579. doi: 10.1002/ijc.34459. Epub 2023 Mar 13. PMID: 36752580 or Yun C, Dashwood WM, Li L, Yin T, Ulusan AM, Shatzer K, Gao S, Ruan KH, Hu M. Acute changes in colonic PGE2 levels as a biomarker of efficacy after treatment of the Pirc (F344/NTac-Apc am1137) rat with celecoxib. Inflamm Res. 2020 Jan;69(1):131-137. doi: 10.1007/s00011-019-01300-5. Epub 2019 Dec 3. PMID: 31797003 and others, which reduced tumor counts or volumes?
Author Response
Reviewer #1
Major comments
Changes of fecal microbiota profile and plasma biomarkers following the administration of an antioxidant oleuropein-rich leaf extract in a rat model mimicking colorectal cancer, uses a rat model of APC driven CRC (also SB tumors) and examines the impact of short-term administration of oleuropein-rich leaf extract on the composition of the fecal microbiota and plasma metabolite composition. This paper is a follow up to a previous study, examining the impact of the extract on tumor apoptosis and iNOS. This follows in vitro studies, looking for synergy between chemotherapeutic agents and the extract in cell line work. The group has further looked at other phenolic rich extracts in this rat model. The work extends to earlier findings showing that the extract alters the fecal microbiota in a way that might favor beneficial bacteria. Plasma metabolites are also significantly altered. However, it is not clear what the next move would be by this group based on the findings.
Authors: We would like to thank the reviewer for the important comments. In this work, by employing a multi-omics approach, integrating 16S rRNA gene sequencing and untargeted metabolomics, we aimed to provide novel insights into the protective role of ORLE against CRC progression. The findings may inform future therapeutic strategies and contribute to the development of targeted interventions leveraging the synergistic interplay between dietary antioxidants, gut microbiota, and host metabolism. We have revised the manuscript to better describe these concepts in each main part.
Detail comments
-The work suggests that short term consumption of the extract altered the fecal microbiota, notably suppressing E. coli/Shig. The authors say Bacteroides was suppressed as well. However, that is not evident in Fig. 2B. Alpha and Beta diversity indexes show marginal changes, more pronounced in wt rats. One of the odd features is the fairly consistent difference in meta-parameters of the microbiota (Fig. S1) between 4 days and 7 days, although not reaching significance. Is there any known reason for this. Did the rats have new cage mates or was the housing otherwise altered when the experiment began?
Authors: Regarding the first point of this comment (The work suggests that short term consumption ……………. However, that is not evident in Fig. 2B), we want to clarify that the alteration in E.coli/Shig refers to a previous work (reference #17) in which an enrichment in Lactobacillus and Paraprevotella (together with a decrease in Escherichia/Shigella and Bacteroides) was found in rats administered Portulaca oleracea polysaccharides, thus suggesting a beneficial effect of that treatment. Therefore, those results were not ours, since, as a matter of fact, we only found that at the class level, Clostridiales were enriched in the faecal bacterial community of ORLE-treated rats. Additionally, at the genus level, an enrichment of Paraprevotella, Anaerotruncus, Oscillibacter and Sporobacter was outlined. In the revised version of the paper we clarified this point better specifying that the reduction in E. coli/Shig refers to a previous paper. See lines 262-266 of the revised manuscript.
Regarding the second point of this comment (One of the odd features ………………), we agree with the referee that this is a notable phenomenon but we do not have any explanation for this. The samples collected at 4 days were a subset of all the samples collected at day 7 (see Fig. 1 B), the rats having the same cage mates for all the duration of the experiment.
-The plasma metabolite study is fairly exhaustive and exhausting to follow. However, since the goal is to establish that the extract does alter the composition, the case is established.
Authors: We modified and better organized the paragraphs related with the plasma metabolomics results and multivariate statistical approaches. Considering the huge amount of information we have focused the attention on the changes of main chemical classes (such as lipids and derivatives) and some biomarker compounds associated with a potential impact of ORLE on plasma metabolome and CRC progression. Please, see revised paragraphs 3.3. and 3.4.
-My major concern is the dosage of the extract. It appears that the dose was somewhat higher than that used by de Bock M, Derraik JG, Brennan CM, Biggs JB, Morgan PE, Hodgkinson SC, Hofman PL, Cutfield WS.(Olive (Olea europaea L.) leaf polyphenols improve insulin sensitivity in middle-aged overweight men: a randomized, placebo-controlled, crossover trial. PLoS One. 2013;8(3):e57622. doi: 10.1371/journal.pone.0057622. Epub 2013 Mar 13. PMID: 23516412; PMCID: PMC3596374.) in their human intervention study. How do you justify the dosage?
Authors: Thank you for the important question. Considering that rats ate about 11 g of diet/day (mean body weight = 300 g), we administered a dose of ORLE of about 100 mg/kg b.w. This dose in rats corresponds (following the human equivalent dose (HED) calculation (Reagan-Shaw et al., Dose translation from animal to human studies revisited. FASEB J. 2008, 22, 659) to 16 mg/kg in humans, a dose not very different from that used in the study by Bock et al., 2013. To properly address this comment, in the revised version of the paper we have added a sentence (Method section, see lines 109-112) explaining the equivalence between rats and human doses.
Additionally, please consider that this is a preliminary work and we used these experimental conditions to better evaluate the impact of ORLE on the plasma metabolomic profile, to extrapolate potential biomarkers associated with CRC progression. We have added a new paragraph stating the limitation of the study to better contextualize this work.
-It is not stated what your next move might be to use the findings from this study. Are you contemplating some sort of fecal transplant studies, without ORLE intervention directly in the study animals-(transplanting from OLRE fed animals into untreated animals) to determine whether the alteration in the microbiota impact tumor growth?
Authors: We thank the Reviewer for this interesting comment. At the moment we do not plan to perform this “expensive” transplantation experiment, but indeed it would be interesting to see whether germ-free animals transplanted with an ORLE modified microbiome had a lower carcinogenesis than rats transplanted with CTR-microbiome.
-Are you going to show any enhanced effect on suppressing tumorigenesis in the PIRC rats using OLRE and some other agents such as those used by Beach R, Hatano Y, Qiao Y, Grady J, Sei S, Mohammed A, Rosenberg DW. Combination of naproxen and a chemically-stable eicosapentaenoic acid analog provide additive tumor protection in Pirc rats. Int J Cancer. 2023 Jun 15;152(12):2567-2579. doi: 10.1002/ijc.34459. Epub 2023 Mar 13. PMID: 36752580 or Yun C, Dashwood WM, Li L, Yin T, Ulusan AM, Shatzer K, Gao S, Ruan KH, Hu M. Acute changes in colonic PGE2 levels as a biomarker of efficacy after treatment of the Pirc (F344/NTac-Apc am1137) rat with celecoxib. Inflamm Res. 2020 Jan;69(1):131-137. doi: 10.1007/s00011-019-01300-5. Epub 2019 Dec 3. PMID: 31797003 and others, which reduced tumor counts or volumes?
Authors: We thank the Reviewer for this observation. Previous studies have already documented a protective effect in the development of colon carcinogenesis by oleuropein administration (Sepporta et al., J. Med. Food 2016, 19, 983–989). We do not know whether our extract (ORLE) would be effective in slowing the carcinogenesis process in PIRC rats, but it would be interesting to investigate in animals aged one month the effect of ORLE in the development of colon carcinogenesis (chemoprevention experiment).
Reviewer 2 Report
The manuscript by Chioccioli et al. describes the impact of Oleuropein treatment on gut microbiome and plasma metabolites. The study design is straightfoward and carried out well. The results are data-heavy and descriptive, which is expected for the type of study. Instead of frequently directing to supplementary materials, it may be better to have a graph/table that describes the key findings.
The manuscript is well-written. A summary figure/table may help to understand the key findings.
Author Response
Reviewer #2
Major comments
-The manuscript by Chioccioli et al. describes the impact of Oleuropein treatment on gut microbiome and plasma metabolites. The study design is straightfoward and carried out well. The results are data-heavy and descriptive, which is expected for the type of study. Instead of frequently directing to supplementary materials, it may be better to have a graph/table that describes the key findings.
Authors: We would like to thank the reviewer for having appreciated this work. The supplementary material consists in different excel sheets, including the metabolomics dataset (resulting from UHPLC-HRMS analysis) containing more than 450 plasma metabolites; the OPLS-DA validation parameters; VIP discriminant metabolites of the OPLS-DA model (a list of 216 plasma metabolites); OPLS-DA models considering the different pairwise comparisons together with the corresponding Pathway Analysis; 5) Exclusive discriminant metabolites resulting from the Venn analysis built considering the VIP biomarkers of each OPLS-DA model built (more than 50 plasma metabolites); and 6) Significant metabolites passing VIP and Volcano plot analyses for the different pairwise comparisons under investigations (more than 100 plasma metabolites). Therefore, we do believe that it's better to keep all this information as supplementary material in order to enhance the readability of the manuscript, considering the huge amount of data and multivariate statistical elaborations.
Detail comments
-The manuscript is well-written. A summary figure/table may help to understand the key findings.
Authors: Thank you for the comment. Given the difficulty in summarizing all the supplementary information in a single table/figure, we have added a Graphical Abstract to better summarize the work and the key findings, showing that ORLE caused an enrichment in bacterial genera with purported beneficial effects such as Paraprevotella, Anaerotruncus, Oscillibacter and Sporobacter.
Reviewer 3 Report
1. To enhance the quality of the manuscript, I would recommend the following:
a) Elaborate on the novelty and significance of the study more explicitly in the introduction, highlighting the knowledge gap it aims to address.
b) Provide a more detailed rationale for the selection of the specific experimental model (PIRC rats) and its relevance to the research question.
c) Discuss the potential limitations of the study, such as sample size or experimental constraints, and suggest future directions to address these limitations.
2. The English usage in the manuscript is generally proficient and appropriate for a scientific publication. However, there are instances of wordiness and unnecessary redundancies that could be improved upon.
Introduction (revised):
In recent decades, there has been a burgeoning interest in exploiting the bioactive compounds present in olive leaves, particularly polyphenols, for their potential applications in the fields of nutraceuticals and biomedicine. Oleuropein (OLE), a secoiridoid phenolic compound abundant in olive leaves, has garnered considerable attention for its putative protective effects against colorectal cancer (CRC). Preclinical studies have suggested that these beneficial activities may be mediated, in part, through modulation of the intestinal microbiota. However, the precise mechanisms underlying this interaction and the subsequent impact on the host metabolome remain elusive.
The present study aims to elucidate the effects of an antioxidant oleuropein-rich leaf extract (ORLE) on the gut microbiota composition and plasma metabolomic profiles in a genetic rat model (PIRC rats) that mimics CRC. By employing a multi-omics approach, integrating 16S rRNA gene sequencing and untargeted metabolomics, this research endeavors to provide novel insights into the protective role of ORLE against CRC progression. The findings may inform future therapeutic strategies and contribute to the development of targeted interventions leveraging the synergistic interplay between dietary antioxidants, gut microbiota, and host metabolism.
Discussion (revised, incorporating the stated disadvantage):
The findings of this study demonstrate a significant modulatory effect of ORLE administration on the gut microbiota composition and plasma metabolomic profiles in PIRC rats, a genetic model mimicking colorectal cancer. Notably, the treatment duration emerged as a critical factor influencing the microbial community structure, with distinct clustering observed between samples collected at different time points (4 days vs. 7 days). This observation underscores the dynamic nature of the gut microbiome and highlights the importance of considering temporal aspects in future investigations.
While the present study provides valuable insights into the potential mechanisms underlying the protective effects of ORLE against CRC progression, it is imperative to acknowledge certain limitations. The sample size, although adequate for the primary analyses, may restrict the statistical power and generalizability of the findings. Additionally, the lack of functional characterization of the observed microbial shifts and their direct impact on host physiology warrants further exploration. Future studies should consider incorporating multi-omics approaches, such as metatranscriptomics and metabolomics of fecal samples, to elucidate the functional implications of the altered gut microbiome.
Another potential limitation lies in the inability to discern the specific contributions of individual phenolic compounds present in the ORLE extract. While oleuropein is the predominant component, the synergistic or antagonistic effects of other phenolics cannot be ruled out. Targeted interventions with purified compounds or fractionated extracts may provide more refined insights into the structure-activity relationships governing the observed biological effects.
Furthermore, the translational potential of these findings to human populations remains to be determined. Validation in clinical cohorts, accounting for inter-individual variability in gut microbiome composition and dietary patterns, is crucial for assessing the feasibility of ORLE supplementation as an adjuvant therapy in CRC management.
Despite these limitations, the present study contributes to the growing body of evidence supporting the chemopreventive potential of olive leaf polyphenols against colorectal cancer. The observed modulation of the gut microbiome and host metabolome by ORLE treatment provides a mechanistic foundation for future investigations aimed at developing targeted interventions that leverage the intricate interplay between dietary antioxidants, gut microbial communities, and host metabolism in the context of chronic diseases such as cancer.
Author Response
Reviewer #3
Major comments
- To enhance the quality of the manuscript, I would recommend the following:
- a) Elaborate on the novelty and significance of the study more explicitly in the introduction, highlighting the knowledge gap it aims to address.
Authors: The Introduction section has been revised, accordingly. Please see lines 38-40, 48-53, 71-83 of the revised manuscript.
- b) Provide a more detailed rationale for the selection of the specific experimental model (PIRC rats) and its relevance to the research question.
Authors: In this work, we used a multiomics approach to evaluate plasma metabolite and microbiota modulations in PIRC rats carrying a heterozygous germline mutation in the Apc gene. The APC mutation is the first event triggering colon carcinogenesis both in the sporadic cases and in familial adenomatous polyposis (FAP) syndrome, a hereditary form of colon cancer [18]. Accordingly, PIRC rats spontaneously develop multiple tumors in the colon and small intestine, thus standing as a robust model to study the protective effect of ORLE, derived from olive leaves, on colon cancer progression. We improved the Introduction section to better explain the relevance of PIRC rats to the research question.
- c) Discuss the potential limitations of the study, such as sample size or experimental constraints, and suggest future directions to address these limitations.
Authors: The Results and Discussion section has been revised and improved, accordingly. We have added a new paragraph to highlight these important points, please see lines 531-562 of the revised manuscript.
- The English usage in the manuscript is generally proficient and appropriate for a scientific publication. However, there are instances of wordiness and unnecessary redundancies that could be improved upon.
Authors: Thank you very much for the comment. We carefully revised the English style and grammar to avoid redundancies.
Detail comments
-Elaborate on the novelty and significance of the study more explicitly in the introduction, highlighting the knowledge gap it aims to address.
Introduction (revised/suggested):
In recent decades, there has been a burgeoning interest in exploiting the bioactive compounds present in olive leaves, particularly polyphenols, for their potential applications in the fields of nutraceuticals and biomedicine. Oleuropein (OLE), a secoiridoid phenolic compound abundant in olive leaves, has garnered considerable attention for its putative protective effects against colorectal cancer (CRC). Preclinical studies have suggested that these beneficial activities may be mediated, in part, through modulation of the intestinal microbiota. However, the precise mechanisms underlying this interaction and the subsequent impact on the host metabolome remain elusive.
The present study aims to elucidate the effects of an antioxidant oleuropein-rich leaf extract (ORLE) on the gut microbiota composition and plasma metabolomic profiles in a genetic rat model (PIRC rats) that mimics CRC. By employing a multi-omics approach, integrating 16S rRNA gene sequencing and untargeted metabolomics, this research endeavors to provide novel insights into the protective role of ORLE against CRC progression. The findings may inform future therapeutic strategies and contribute to the development of targeted interventions leveraging the synergistic interplay between dietary antioxidants, gut microbiota, and host metabolism.
Authors: The Introduction section has been revised, accordingly.
-Discuss the potential limitations of the study, such as sample size or experimental constraints, and suggest future directions to address these limitations
Discussion (revised, incorporating the stated disadvantage):
The findings of this study demonstrate a significant modulatory effect of ORLE administration on the gut microbiota composition and plasma metabolomic profiles in PIRC rats, a genetic model mimicking colorectal cancer. Notably, the treatment duration emerged as a critical factor influencing the microbial community structure, with distinct clustering observed between samples collected at different time points (4 days vs. 7 days). This observation underscores the dynamic nature of the gut microbiome and highlights the importance of considering temporal aspects in future investigations.
While the present study provides valuable insights into the potential mechanisms underlying the protective effects of ORLE against CRC progression, it is imperative to acknowledge certain limitations. The sample size, although adequate for the primary analyses, may restrict the statistical power and generalizability of the findings. Additionally, the lack of functional characterization of the observed microbial shifts and their direct impact on host physiology warrants further exploration. Future studies should consider incorporating multi-omics approaches, such as metatranscriptomics and metabolomics of fecal samples, to elucidate the functional implications of the altered gut microbiome.
Another potential limitation lies in the inability to discern the specific contributions of individual phenolic compounds present in the ORLE extract. While oleuropein is the predominant component, the synergistic or antagonistic effects of other phenolics cannot be ruled out. Targeted interventions with purified compounds or fractionated extracts may provide more refined insights into the structure-activity relationships governing the observed biological effects.
Furthermore, the translational potential of these findings to human populations remains to be determined. Validation in clinical cohorts, accounting for inter-individual variability in gut microbiome composition and dietary patterns, is crucial for assessing the feasibility of ORLE supplementation as an adjuvant therapy in CRC management.
Despite these limitations, the present study contributes to the growing body of evidence supporting the chemopreventive potential of olive leaf polyphenols against colorectal cancer. The observed modulation of the gut microbiome and host metabolome by ORLE treatment provides a mechanistic foundation for future investigations aimed at developing targeted interventions that leverage the intricate interplay between dietary antioxidants, gut microbial communities, and host metabolism in the context of chronic diseases such as cancer.
Authors: The Results and Discussion section has been revised and improved, accordingly.
Round 2
Reviewer 1 Report
The paper as currently written contributes to the knowledge base of polyphenols in CRC.
The revisions render the paper suitable for publication.
Author Response
Thank you
Reviewer 3 Report
Dear colleagues,
As a seasoned researcher with decades of experience in the field of medicine, allow me to provide some respectful feedback and recommendations regarding this interesting study.
1. The study design and results presented appear rigorous and well-executed. Some additional discussion contextualizing the clinical relevance and translational potential of modulating gut microbiota and plasma biomarkers through supplementation could strengthen the work.
2. Including a brief overview of the pathogenesis and risk factors for colorectal cancer would help orient non-expert readers. Correlating particular microbial and metabolic changes to disease progression markers could deepen the insights.
3. Considering exploring opportunities to validate some findings through targeted experimental work. For example, conducting selective microbiota depletion/modulation to identify causative relationships.
The language used was consistently clear, concise and scientifically precise as expected for a peer-reviewed academic journal. Terms were well-defined on first use and acronyms explained. For non-native English speakers, subject-verb agreement and the occasional preposition usage could be improved but did not detract from comprehension.
Among its strengths, the study demonstrated rigorous methodology and generated intriguing multi-omic data investigating an understudied area. A limitation was the lack of human clinical correlations due to the early-stage research context.
In closing, I commend your diligent work advancing our understanding of polyphenol metabolism and microbiome impacts. Multi-disciplinary collaboration holds promise for developing novel prevention and treatment strategies. Stay encouraged in following the facts wherever they lead and pushing scientific boundaries for the benefit of humanity.
Dear colleagues,
As a seasoned researcher with decades of experience in the field of medicine, allow me to provide some respectful feedback and recommendations regarding this interesting study.
1. The study design and results presented appear rigorous and well-executed. Some additional discussion contextualizing the clinical relevance and translational potential of modulating gut microbiota and plasma biomarkers through supplementation could strengthen the work.
2. Including a brief overview of the pathogenesis and risk factors for colorectal cancer would help orient non-expert readers. Correlating particular microbial and metabolic changes to disease progression markers could deepen the insights.
3. Considering exploring opportunities to validate some findings through targeted experimental work. For example, conducting selective microbiota depletion/modulation to identify causative relationships.
The language used was consistently clear, concise and scientifically precise as expected for a peer-reviewed academic journal. Terms were well-defined on first use and acronyms explained. For non-native English speakers, subject-verb agreement and the occasional preposition usage could be improved but did not detract from comprehension.
Among its strengths, the study demonstrated rigorous methodology and generated intriguing multi-omic data investigating an understudied area. A limitation was the lack of human clinical correlations due to the early-stage research context.
In closing, I commend your diligent work advancing our understanding of polyphenol metabolism and microbiome impacts. Multi-disciplinary collaboration holds promise for developing novel prevention and treatment strategies. Stay encouraged in following the facts wherever they lead and pushing scientific boundaries for the benefit of humanity.
Author Response
Reviewer #3
Dear colleagues,
As a seasoned researcher with decades of experience in the field of medicine, allow me to provide some respectful feedback and recommendations regarding this interesting study.
- The study design and results presented appear rigorous and well-executed. Some additional discussion contextualizing the clinical relevance and translational potential of modulating gut microbiota and plasma biomarkers through supplementation could strengthen the work.
Authors: The authors thank the Reviewer for this observation. In the revised version of the paper according to the previous comments by this referee, we added a sentence (see lines 562-566) saying that “ the translational potential of these findings to human populations remains to be determined. Validation in clinical cohorts, accounting for inter-individual variability in gut microbiome composition and dietary patterns, is crucial for assessing the feasibility of ORLE supplementation as an adjuvant therapy in CRC management.” We are at an early stage of this research thus, further comments on the clinical relevance should be supported by additional studies.
- Including a brief overview of the pathogenesis and risk factors for colorectal cancer would help orient non-expert readers. Correlating particular microbial and metabolic changes to disease progression markers could deepen the insights.
Authors: The authors thank the reviewer for this suggestion. Additional information on risk factors for CRC and the role of gut microbiota in CRC has been added to the introduction section see lines 50-54.
- Considering exploring opportunities to validate some findings through targeted experimental work. For example, conducting selective microbiota depletion/modulation to identify causative relationships.
Authors: The authors thank the Reviewer for this interesting comment. A possibility could be to perform a faecal microbiota transplantation from rats treated or not with ORLE. At the moment we do not plan to perform this “expensive” transplantation experiment, but it would be interesting to see whether germ-free animals transplanted with an ORLE modified microbiome had a lower carcinogenesis compared to rats transplanted with CTR-microbiome.
The language used was consistently clear, concise and scientifically precise as expected for a peer-reviewed academic journal. Terms were well-defined on first use and acronyms explained. For non-native English speakers, subject-verb agreement and the occasional preposition usage could be improved but did not detract from comprehension.
Authors: We corrected, as possible, the few errors present.
Among its strengths, the study demonstrated rigorous methodology and generated intriguing multi-omic data investigating an understudied area. A limitation was the lack of human clinical correlations due to the early-stage research context.
Authors: Thanks to the reviewer for this observation. We agree with the Reviewer for the lack of human clinical correlations. The aim of our work was to investigate if ORLE extract was able to modulate and ameliorate some tumour-promoting mechanisms connected to CRC in an experimental model. Future studies could be set up to study whether this protective effect could be reached also in CRC patients, but as stated above, it is premature to speculate about a possible human transability approach.
In closing, I commend your diligent work advancing our understanding of polyphenol metabolism and microbiome impacts. Multi-disciplinary collaboration holds promise for developing novel prevention and treatment strategies. Stay encouraged in following the facts wherever they lead and pushing scientific boundaries for the benefit of humanity.
Authors: The authors thanks the Reviewer for this comment.